# Isolation and Functional Characterization of a Green-Tissue Promoter in Japonica Rice (*Oryza sativa* subsp. *Japonica*)

**DOI:** 10.3390/biology11081092

**Published:** 2022-07-22

**Authors:** Mi Lin, Jingwan Yan, Muhammad Moaaz Ali, Shaojuan Wang, Shengnan Tian, Faxing Chen, Zhimin Lin

**Affiliations:** 1Fujian Academy of Agricultural Sciences Biotechnology Institute, Fuzhou 350003, China; linmi1009@163.com (M.L.); yjw@fjage.org (J.Y.); 2College of Horticulture, Fujian Agriculture and Forestry University, Fuzhou 350002, China; muhammadmoaazali@yahoo.com (M.M.A.); wsj975620wsj@163.com (S.W.); tsl0224@163.com (S.T.)

**Keywords:** promoter, green-tissue specific expression, cloning, *agrobacterium*, RT-PCR

## Abstract

**Simple Summary:**

Transgenic applications have largely focused on constitutive promoters in plants. However, strong and continuous over-expression of certain genes may be redundant and even harmful to plant growth. Thus, tissue-specific promoters are the most suitable for regulating target gene expression. Although several tissue-specific promoters have been identified, the regulatory mechanism of tissue-specific gene expression remains unclear. By a series of GUS staining of 5′ and 3′ deletions, we uncover tissue-specific *cis*-acting elements in *GSX7R*, including ten light-responsive elements. The results reveal that *GSX7R* is a reverse green tissue-specific promoter, except in endosperm. In contrast, strong tissue-specific promoters that can be used for rice improvements are limited. In this study, we successfully showed that the *GSX7R* promoter can drive the *Cry1Ab* gene to resistant rice yellow stem borer. In addition, our study demonstrates an effective promoter to drive foreign genes for crop improvement.

**Abstract:**

Plant promoters play a vital role in the initiation and regulation of gene transcription. In this study, a rice protein/gene of unknown expression, named *Os8GSX7*, was gained from a rice T-DNA capture line. The semi-quantitative RT-PCR analysis showed that the gene was only expressed in root, glume, and flower, but not in stem, leaf, embryo, and endosperm of japonica rice. The GUS activity analysis of the *GSX7R* promoter showed that it was a reverse green tissue expression promoter, except in endosperm. The forward promoter of *GSX7* cannot normally drive the expression of the foreign *GUS* gene, while the reverse promoter of *GSX7* is a green tissue-specific expression promoter, which can drive the expression of the foreign *GUS* gene. The region from −2097 to −1543 bp was the key region for controlling the green tissue-specific expression. The regulatory sequences with different lengths from the 2097 bp reverse sequence from the upstream region of the *Os8GSX7* were fused with the *GUS* reporter gene and stably expressed in rice. Furthermore, transgenic rice plants carrying Cry1Ab encoding *Bacillus thuringiensis* endotoxin, regulated by *GSX7R*, were resistant to yellow stem borer. The analysis suggested that 10 light responsive elements of tissue-specific expression were found, including ACE, Box4, CAT-box, G-Box, G-box, GATA motif, GC motif, I-box, Sp1, and chs-unit1 M1. In addition, the results of 5′ and 3′ deletions further speculated that ACE and I-box may be the key elements for determining the green tissue-specific expression of *GSX7R* promoter.

## 1. Introduction

Promoters are the key regulators of plant genetic engineering, during the process of gene transcription [1]. They are mainly divided into three types, i.e., constitutive, inducible, and tissue-specific. Constitutive promoters are usually used to analyze gene function and character modification, which can cause foreign genes to obtain a stable expression level in plants [2]. There are three most commonly used constitutive promoters in genetic engineering, such as 35S, actin, and ubiquitin [3,4,5]. However, a large number of efficient promoters which are continuously driving the expression of foreign genes are not only unnecessary for plant growth, but also consume a lot of nutrients in plants, so far as to have a negative impact on plant growth and development [6]. The transcription of constitutive promoters often inhibits plant growth [7]. Tissue-specific promoters are generally a 5′-UTR of genes expressed in specific tissues, organs, or specific development stages, which can effectively avoid negative effects on plant growth and development [1]. Although some tissue-specific expression promoters have been cloned, mostly including root [8], leaf sheath [9], phloem [10], pollen grain [11], embryo [12], endosperm [13] and green tissue [14,15,16,17], but only a small part of *cis*-acting regulatory elements are recognized [18]. In addition, the cloned green tissue-specific expression promoters are mainly some genes closely related to light induction and photosynthetic system, which participate in plant photosynthesis and realize the function of converting light energy into sugar [19], such as *RBCs* [20], *DX1* [14], *D540* [21]. Isolation and identification of rice green tissue specific expression promoters related to the non-photosynthetic system are helpful to avoiding the influence of light effect, and have a wider range of applications.

The broader expression pattern mainly focuses on the forward promoter [22]. The reverse promoter mainly means that its forward promoter cannot drive the expression of foreign genes. However, a few studies are reported on plant reverse promoters, most of the studies are focused on bi-directional promoters, such as *BEAP* [23], *Cab1/2* [24] and *BDDP* [25]. Rice is one of the most important food crops in the world [26,27]. The application of strong specific promoters is very important for rice genetic improvement [28]. In this study, a green tissue-specific promoter was obtained through the screening of the T-DNA capture line. The gene information and regulation of the promoter have not been unknown. The reverse promoter fragment 2097 bp upstream of the *Os8GSX7* gene was obtained by the PCR cloning method. Furthermore, the 5′ and 3′ end deletion of the reverse promoter *GSX7R* was analyzed, and the corresponding plant expression vectors were constructed. Through *agrobacterium*-mediated genetic transformation of rice and GUS tissue staining, we characterized a core region of green-specific control of gene expression. It also provides a reference theoretical basis for the expression and regulation function of the unknown protein. More importantly, due to its strong tissue-specific expression, it can provide an application basis for transgenic safety in the future.

## 2. Materials and Methods

### 2.1. Experimental Materials

Japonica rice (*Oryza sativa* subsp. Japonica), a fine variety was used as a receptor. An independently constructed rice T-DNA insertion mutant library promoter (Figure 1), *Escherichia coli* strain DH5α, and *Agrobacterium tumefaciens* strain LBA4404 were preserved in the laboratory. The promoter expression verification vector pCXGUS-P was constructed and provided by Dr. Chen Songbiao [29].

### 2.2. Expression Analysis of Candidate Genes in Rice Organs

An unknown expression gene (LOC_Os01g35580) was obtained by T-DNA capture line, named *Os8GSX7*, and its promoter was reverse promoter GSX7R. Using RNA extraction kit (Tiangen Biotech Co., Ltd., Beijing, China), total RNA was extracted from various tissues and organs of japonica rice, including root, stem, leaf, leaf sheath, glume, flower, embryo, and endosperm. A total of 2 µg sample of RNA was reverse transcribed into cDNA (Tiangen Biotech Co., Ltd., Beijing, China).

Using the cDNA obtained from the above reverse transcription as the template, RT-PCR semi-quantitative analysis was carried out, and primers were designed using primer 6.0 software to detect the expression of genes (Table 1). The PCR reaction conditions were: 94 °C denaturation for 3 min, 94 °C 15 s, 58.5 °C 15 s, 72 °C 30 s, 28 cycles, extension at 72 °C for 5 min, and storage at 4 °C. *Os8GSX7* and *GUS* genes were amplified with rice actin gene as internal reference. The amplified fragments were 114 bp and 203 bp respectively. PCR products were analyzed by 1.5% agarose gel electrophoresis.

### 2.3. Promoter Cloning and Cis-Acting Element Analysis

Using the 2097 bp sequence truncated from the upstream region of the coding gene sequence ATG of japonica rice, primer 6.0 software was used for primer design (Table 1). Primer design PCR method was used to construct the 5′ and 3′ deletion promoters of forward promoter GSX7F and reverse promoter GSX7R series.

The 5′ and 3′ deletion fragments were cloned into the promoter cloning vector pCXGUS-P by TA cloning through PCR (Figure 2). *Agrobacterium tumefaciens* LBA4404 was transformed by electric shock, and the mature embryo callus of rice was transformed by *Agrobacterium tumefaciens* and introduced into Nipponbare.

### 2.4. Construction of Plant Expression Vector

PCR system (50 µL): 2× Phanta Max Buffer 25 µL, dNTP Mix (10 mM each) 1 µL, template DNA (100 ng/µL), upstream primer (10 µM) 2 µL, downstream primer (10 µM) 2 µL, Phanta Max Super-Fidelity DNA Polymerase 1 µL, insufficient supplement ddH_2_O to 50 µL. PCR double stress condition: 95 °C 3 min; 95 °C 15 s, 60 °C 15 s, 72 °C for 2 min 30 s, 35 cycles; 72 °C for 5 min. After adding a tail to the PCR product, it was constructed on the skeleton vector pCXGUS-P digested by *Xcm*I and sent to Xiamen Boson Biotech Co., Ltd., Xiamen, China for sequencing verification.

The p1300GSAb vector was constructed by one step cloning kit (Vazyme Biotech, code: C112). The pCAMBIA1300 vector was designed by enzyme digestion by *Hin*dIII and *Eco*RI. GSX7R promoter, Cry1Ab and Tnos were amplified by PCR, and then connected to pCAMBIA1300.

### 2.5. Analysis of Genetic Transmission and Transgenic Detection

After mechanical shelling of Nipponbare seeds, they were pretreated with 75% ethanol for 2 min, soaked in 3% sodium hypochlorite for 20 min, rinsed with sterile water for 4–5 times, and placed on an N6 medium containing 3 mg/L 2,4-D, dark cultured at 25 °C to induce callus, transformed by Agrobacterium-mediated rice genetic transformation method, and regenerated rice plants [30]. The PCR amplification product of HPT gene was 832 bp [31].

### 2.6. GUS Histochemical Staining

GUS chemical staining was performed on different tissues and organs of transgenic positive plants according to the previously described method [14]. The roots, stems, leaves, leaf sheaths, glumes, and seeds of positive transgenic plants were cut into appropriate sizes with blades, and put into EP tubes. An appropriate amount of GUS dye was added to make them fully colored [GUS dye composition: 50 mM sodium phosphate (pH 7.0), 10 mmol Na_2_EDTA, 0.1% Triton X-100, 1 mg/mL x-gluc, 100 μg/mL chloramphenicol, 1 mM potassium ferricyanide, 1 mm/L potassium ferrocyanide and 20% methanol]. The treating samples were placed at 37 °C overnight. After dyeing, it was decolorized with absolute ethanol and 70% (*v/v*) ethanol for about 1 h. At least 10 repetitions were set for each transformation event, and the photos were taken with a stereomicroscope (OLYMPUS-SZ61, Olympus Corporation, Tokyo, Japan).

## 3. Results

### 3.1. The Expression Pattern Analysis of Candidate Gene Os8GSX7 in Rice

The cDNA was obtained by double transcription of total RNA from different tissues and organs of japonica rice. It was used as a template for PCR double correspondence. The expression pattern of candidate gene *Os8GSX7* was analyzed by semi-quantitative RT-PCR. The results showed that the *Os8GSX7* gene was only expressed in roots, glumes, and flowers (Figure 3A). The reverse promoter GSX7R-2097 derived the expression of exogenous *GUS* gene in the green tissue of rice, stem, leaf, glume, flower, and embryo, as well as in root but not in the endosperm (Figure 3B).

### 3.2. Isolation and Cloning of the Full-Length Promoter GSX7R and Its Deletion Fragment

Using the genomic DNA of Nipponbare rice as the template, a reverse 2097 bp promoter fragment from the upstream region of the gene was cloned by PCR and named GSX7R. Using pCXGUS-GSX7R as a template, other 5′ and 3′ deletion fragments were cloned respectively (Figure 4). The fragment was connected to the plant expression vector by the TA cloning method, and *Agrobacterium tumefaciens* was transformed into rice callus. All transgenic plants with positive fragments were identified by hygromycin marker gene detection PCR and planted in a greenhouse.

### 3.3. Bioinformatics Analysis of Reverse Promoter GSX7R Sequence

Through the promoter prediction software plantCARE *cis*-acting element database, the promoter GSX7R sequence of 2097 bp reverse fragment upstream of the *Os8GSX7* gene was analyzed. There were mainly 46 elements. In addition to the core elements of the CAAT box and TATA box of the promoter, it was found that there were 10 potential key *cis*-regulatory elements specifically expressed by green tissue (Appendix A, Figure 5), including ACE, Box4, CAT box, G-Box, G-box, GATA motif, GC motif, I-box, Sp1, and chs-unit1 M1 components.

### 3.4. Molecular Detection of Transgenic Rice

The screening marker hygromycin gene (*hpt*) was used to detect the genes of rice-positive transgenic plants with different vector skeletons. The results showed that the positive progeny plants were isolated from different transformed plants (Figure 6). The size of the hygromycin gene was in line with the expected 832 bp.

### 3.5. GUS Activity Analysis of GSX7R and Truncated Promoter

The roots, stems, leaves, leaf sheaths, glumes, and seeds of transgenic positive plants were selected for GUS staining analysis. The *GUS* gene driven by constitutive act promoter was positive. The results showed that the forward *GSX7* full-length promoter could not start GUS protein expression, while the reverse *GSX7R* full-length promoter could start GUS protein expression, so *GSX7R* was a reverse expression promoter. The *GSX7R* promoter was positive in other green tissues except for the expression of GUS protein in the endosperm (Figure 7a), which further confirmed that the reverse promoter *GSX7R* was a green tissue-specific expression promoter.

The results showed that GSX7R-1765, GSX7R-1198, and GSX7R-554 deleted at the 5′ ends could drive the expression of the *GUS* gene in the stem, leaf, leaf sheath, and glume, but not in root and seed. However, GSX7R-1543 and GSX7R-1212 with 3′ deletions could not drive the expression of the *GUS* reporter gene. Therefore, the promoter element at the 3′ ends is very critical to the function of the promoter. To determine the strength of the seven *GSX7R* promoter variants, the quantitative expression of *GUS* mRNA in leaves was determined. Compared to the histochemical staining of GUS, the levels of *GUS* transcript were designated as “no expression” in *GSX7* transformed rice. Compared with the *GUS* expression driven by only the *Actin* promoter of p1300AGS, GSX7R-2097 and GSX7R-1765 had significantly strong expression (Figure 7b). These results demonstrated that the *GSX7R* promoter might play an important role in the regulation of gene expression in the green tissue of plants.

### 3.6. Evaluation of the Ability of GSX7R Promoter to Drive Foreign Genes

To evaluate the ability of *GSX7R* promoter to drive foreign genes, a transgenic vector was constructed (Figure 8a) and insect resistance was carried between transgenic and wild-type (WT) plants in the greenhouse paddy field. The results showed that WT plants were severely damaged by yellow stem borer (Figure 8b(B,D)), whereas the transgenic plants grew well (Figure 8b(A,C)). It demonstrated that the *GSX7R* promoter had the strong ability to drive the expression of foreign genes.

## 4. Discussion

The plant promoters have significant importance with respect to plant biotechnology [32]. The precise regulation of genes mainly depends on the promoters. The characteristics of promoter elements basically determine the spatio-temporal expression and transcription level of the gene [33]. Generally, there are significant differences in promoters within and between genomes, which shows that some promoters contain specific types of bases, while others have diverse and complex sequence characteristics [34]. In this study, the surface characteristics of a 2097 bp reverse promoter upstream of the *Os8GSX7* (LOC_Os01g35580) gene in rice were studied. The forward promoter could drive the expression of rice endogenous gene *Os8GSX7* in roots, glumes, and flowers, but it could not drive the expression of the *GUS* reporter gene. The reverse promoter *GSX7R* had high intensity and could drive the specific expression of the *GUS* reporter gene in rice green tissue (Figure 7a). Compared with the constitutive promoter *p1300AGS*, except that the expression was not detected in endosperm and was detected in the root, other green tissue parts were expressed, including stem, leaf, leaf sheath, glume, and embryo.

In order to further study the function of green tissue-specific expression *cis*-acting elements, we analyzed the 5′ and 3′ deletion truncation of *GSX7R* promoter, constructed seven different full-length and deletion promoter expression vectors, transformed them into rice, and obtained their transgenic positive plants by hygromycin resistance screening and PCR detection of resistance genes. In addition to the promoter core element TATA box, CAAT box, and GC motif that enhance transcription efficiency, 10 elements closely related to the specific expression of light-response were found e.g., ACE, Box4, CAT box, G-Box, G-box, GATA motif, GC motif, I-box, Sp1, and chs-unit1 M1. From the distribution position, they were mainly distributed in the −1021 to −1621 region of the promoter. When *GSX7R* promoter 5′ was deleted to −1765, −1198, and −554, *GUS* expression was not detected in roots and embryos, but was still strongly expressed in stems, leaves, leaf sheaths, and glumes. When the 3′ ends were deleted to −1212 and −1543, the promoter lost the activity of driving foreign genes and *GUS* expression was not detected (Figure 7a). Combined with the *GUS* mRNA expression of the 3′ end fragment (Figure 7b), it can be shown that ACE, SP1, GC motif, GATA motif, I-box, and Box4 elements are very important for the tissue-specific expression of promoter *GSX7R*. Analysis together with the results of 5′ end fragment deletion showed that ACE and I-box may play a decisive role in the expression of green tissue promoter *GSX7R*. Previous studies have shown that the ACE element in the promoter is regulated by HY5, and the lack of HY5 will widely reduce the accumulation of other photosystem proteins except for PSII protein [35]. I-box elements mainly regulate the activity of promoters in leaves, not fruits, and can control light regulatory genes [36,37]. In addition, I-box and G-box elements act as light response enhancers of CMA5 activity in plants [38,39].

In addition to light response-related elements, the promoter *GSX7R* also had meristem element (CAT box), MYBHv1 binding site (CCAAT box), MYB transcription element (MBS, MYB, MYB recognition site, MYB-like, and MYB binding site), seed-specific expression element (RY-element), auxin response element (TGA element), jasmonic acid response element (CGTCA motif), etc. Studies have shown that genes with MYB *cis*-acting elements can combine with jasmonic acid response to improve cold tolerance in rice [40]. In addition, the rice *OsMYB4* transcription factor can directly or indirectly regulate the tolerance of target genes by interacting with transcription factors such as the CCAAT box and MYB [41]. The combination of MYB binding site element and TIMYB2r-1 protein can improve the disease resistance of wheat [42]. Therefore, the functional study of these elements may provide a research direction for revealing the unknown functional genes of *Os8GSX7* (LOC_Os01g35580).

## 5. Conclusions

In this study, a rice reverse promoter (Os*CGX7R*) was obtained through T-DNA capture line screening. The expression intensity of the reverse promoter was higher than that of the forward promoter, and the reverse promoter was a green tissue efficient expression promoter. Compared with the green tissue-specific expression promoter related to the traditional photosynthetic system, it can avoid the influence of light regulation and has wider application. The results of this study can provide a new basis for the application and development of green tissue-specific expression promoters in transgenic safety.

## Figures and Tables

**Figure 1 biology-11-01092-f001:**
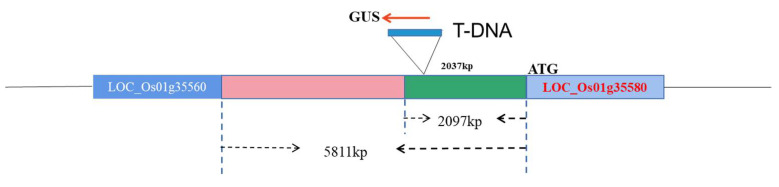
A schematic diagram showing the insertion position of T-DNA.

**Figure 2 biology-11-01092-f002:**
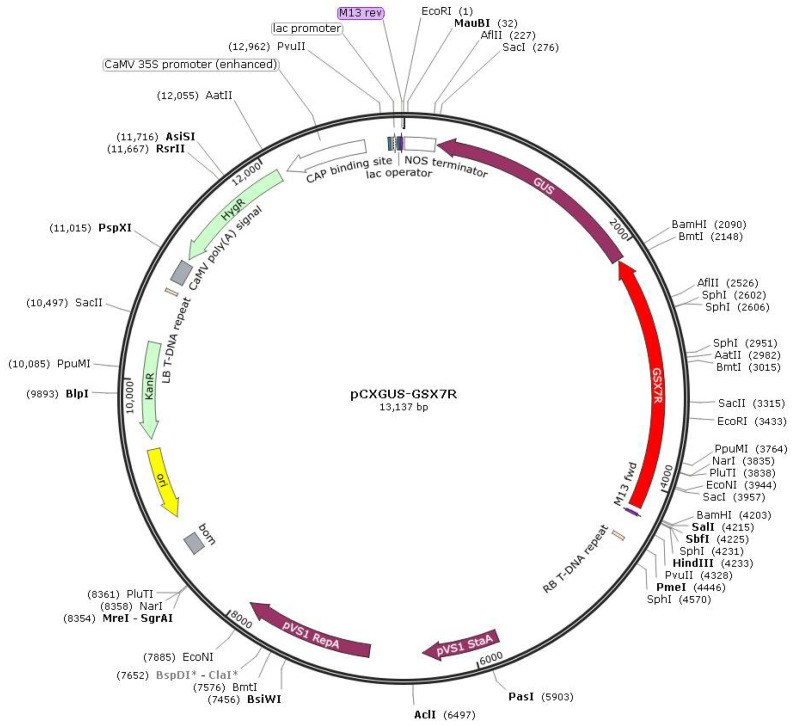
Schematic diagram of the T-DNA region of binary vector pCXGUS-P with GSX7R or GSX7R deletions.

**Figure 3 biology-11-01092-f003:**
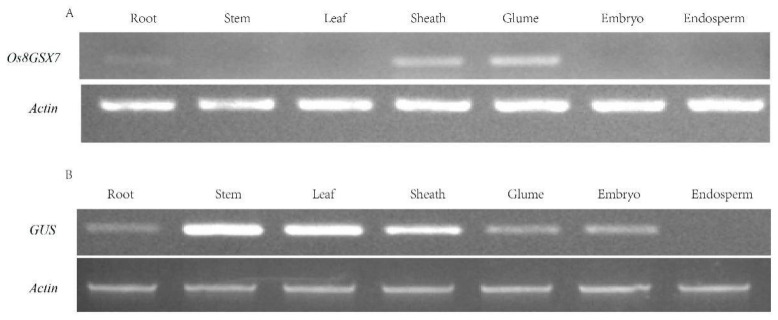
Expression profiles of *Os8GSX7* and *GUS* by RT-PCR analysis. (**A**) Expression profiles of *Os8GSX7* gene by RT-PCR analysis in the wild type; (**B**) Expression profiles of *GUS* gene in transgenic positive lines of GSX7R-2097 by RT-PCR analysis. The rice Actin gene was used as an internal control.

**Figure 4 biology-11-01092-f004:**
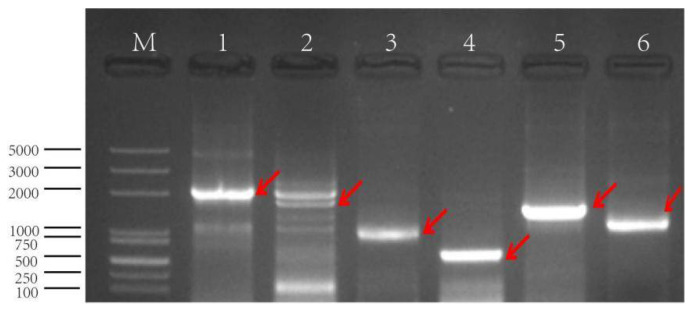
The PCR amplification of promoter *GSX7R* and its deletions. M: M5 DL2000 plus DNA marker, 1: GSX7R-2097, 2: GSX7R-1765, 3: GSX7R-1198, 4: GSX7R-554, 5: GSX7R-1543, 6: GSX7R-1212. The amplified bands are indicated by red arrows.

**Figure 5 biology-11-01092-f005:**
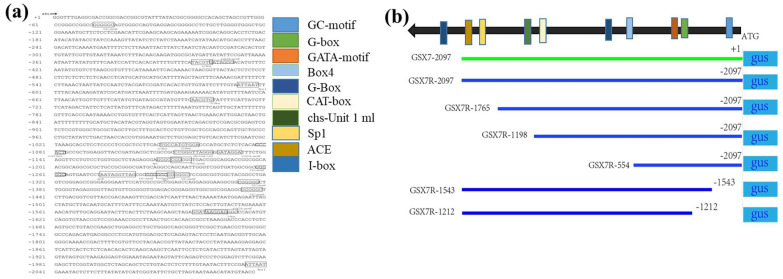
The location of putative *cis*-acting elements in *GSX7R* predicted by the PlantCARE database and schematic diagrams of promoter deletion constructs. (**a**) Putative *cis*-acting elements in *GSX7R*, The reverse 5′-region of the *Os8GSX7* gene containing the 2097 bp promoter sequence from the translational start site. The transcription initiation site is defined as +1. The GC-motif, G-box, and other key cis-acting elements are underlined with and indicated by black frame as shown in the legend and different colors outside. The position of each element is also indicated by schematic diagrams; (**b**) The schematic diagrams of the truncated *GSX7R* constructs. The numbers to the left of these diagrams indicate the position of the 5′-deletion or 3′-deletion.

**Figure 6 biology-11-01092-f006:**
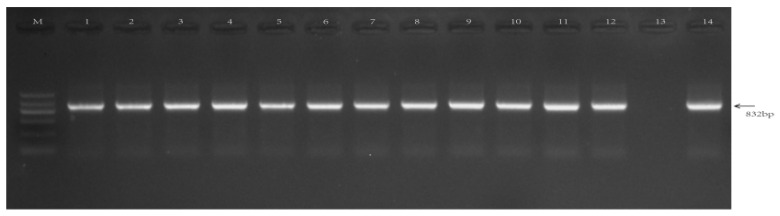
PCR analysis of *hpt* in different transgenic rice. M: M5 Marker II DNA Marker, 1–2: GSX7R-2097; 3–4: GSX7R-1765, 5–6: GSX7R-1198, 7–8: GSX7R-554, 9–10: GSX7R-1543, 11–12: GSX7R-1212, 13: non-transgenic rice, 14: Positive plasmid control.

**Figure 7 biology-11-01092-f007:**
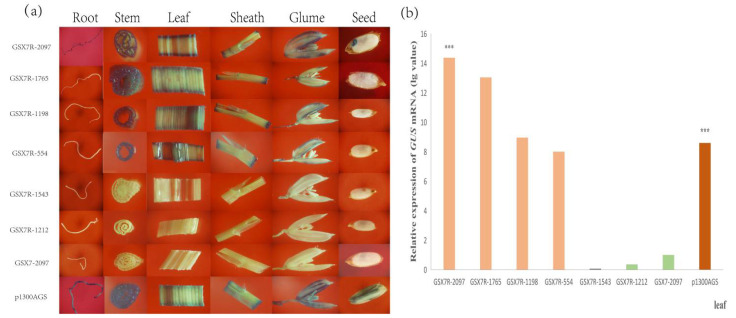
GUS staining and quantification of *GUS* mRNA transcript of transgenic rice containing different 5′ and 3′-deletion. (**a**) Histochemical analysis of transgenic rice plants containing various *GSX7R* promoter fragment/*GUS* fusions in different tissues; (**b**) *GUS* mRNA expression in the leaf of various rice transformants. Quantitative qRT-PCR analysis of *GUS* transcripts from the leaf in transgenic rice plants. Data are shown as mean ± SD (n = 3), Student’s *t*-test. *** indicates highly significant differences from all tested tissues for each transgenic plant. Relative expression was calculated as lg_2_^−ΔΔCt^.

**Figure 8 biology-11-01092-f008:**
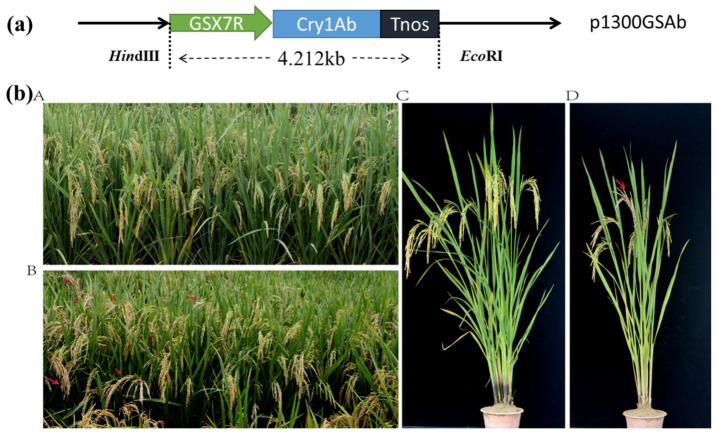
Schematic diagram of vector construction and bioassays in the field. (**a**) GSX7R:Cry1Ab:p1300GSAb plant transformation vector containing the *Cry1Ab* gene driven by the *GSX7R* promoter; (**b**) (**A**,**C**) Transgenic plants without any symptom of infestation in the whole-plant bioassay; (**B**,**D**) Wild-type plants showed white heads in the whole-plant bioassay. Red arrow indicates damage to YSB.

**Table 1 biology-11-01092-t001:** PCR primers used in the study.

Primer Name	Sequence(5′-3′)	Usage
GSX7-2097F	GGTTACATATGTTTATTACTAAGC	Full-length cloning of forward promoter
GSX7-2097R	GGGTTTGAGGCGACCGGCGACCGGC	
GSX7R-2097F	GGGTTTGAGGCGACCGGCGACCGGC	Full length cloning of reverse promoter
GSX7R3R	GGTTACATATGTTTATTACTAAGCA	
GSX7R-1765F	ATGGCGCATGATTATATTCCGATTA	5′ deletion cloning
GSX7R3R	GGTTACATATGTTTATTACTAAGCA	
GSX7R-1198F	GTCTCCGTGGGCTGCGCTAGCTTGC	5′ deletion cloning
GSX7R3R	GGTTACATATGTTTATTACTAAGCA	
GSX7R-554F	ACTTGTACTTAGAAAATAACATGTT	5′ deletion cloning
GSX7R3R	GGTTACATATGTTTATTACTAAGCA	
GSX7R-2097F	GGGTTTGAGGCGACCGGCGACCGGC	3′ deletion cloning
GSX7R-1543R	GGAGATAGACATTATTTGGAAATGA	
GSX7R-2097F	GGGTTTGAGGCGACCGGCGACCGGC	3′ deletion cloning
GSX7R-1212R	CGCGCCTGCCGTTGCCGCCGGTCCT	
*hpt*F	ACACAGCCATCGGTCCAGA	hpt maker gene identification
*hpt*R	TAGGAGGGCGTGGATATGTC	
*Os8GSX7*F	GCTCGACGCATGCATGGCACAG	RT-PCR
*Os8GSX7*R	GTCCAATATGTGGAATCTGATC	
*GUS*F	GAACTGGCAGACTATCCCGCCGG	RT-PCR
*GUS*R	CCTGCCAGTCAACAGACGCGTGG	
*Actin*F	CATGCTATCCCTCGTCTCG	Reference gene
*Actin*R	CGCACTTCATGATGGAGTTG	

## Data Availability

Not applicable.

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
