# Peer review of "Isolation and Functional Characterization of a Green-Tissue Promoter in Japonica Rice (Oryza sativa subsp. Japonica)"

_biology, 2022, doi:10.3390/biology11081092_

Round 1

Reviewer 1 Report

This manuscript is well written, but I only observe that most of the figures the description is not clear, for example figure 7 a the horizontal and vertical axis is not clear. this needs to improve.

Author Response

This manuscript is well written, but I only observe that most of the figures the description is not clear, for example figure 7 a the horizontal and vertical axis is not clear. this needs to improve.

Response: Respected reviewer, thank you so much for the appreciation of our work. The figures are replaced with more clear ones. The figure 7a has also been improved.

Reviewer 2 Report

1. Since the promoter can drive GUS expression in root, you can't say "Green-Tissue Promoter OsGSX7". You may change to "Green-Tissue preferred..". 

2. Explain what is reverse promoter in the introduction section more clearly.

3. Line 150-151: "The reverse promoter GSX7R-2097 derived the expression of exogenous GUS gene in the green tissue of rice, including root, stem, leaf, glume, flower and embryo, but not in endosperm". I don't think root is green tissue, you should modify it and similar sentences across the manuscript.

4. Fig 4: No. 2 lane has multiple bands, you should indicate which band is the target band.

5. Line 229-230: "Figure 8bB,bD" change to "Figure 8b B&D"; "Figure 8bA,bC" change to " Figure 8b A&C".

6. Line 260: what is the difference of the two "G-Box, G-box"? 

7. Line 283: "genes by interacting with transcription factors such as CCAAT box and MYB". I don't think CCAAT box is a transcription factor. The authors should read the manuscript carefully again and improve the English writing.

Author Response

  1. Since the promoter can drive GUS expression in root, you can't say "Green-Tissue Promoter OsGSX7". You may change to "Green-Tissue preferred..". 

Response: Thank you for the valuable suggestion. The title has been modified.

  1. Explain what is reverse promoter in the introduction section more clearly.

Response: The introduction of reverse promoter has been incorporated in Introduction section (see lines 64-67)

  1. Line 150-151: "The reverse promoter GSX7R-2097 derived the expression of exogenous GUS gene in the green tissue of rice, including root, stem, leaf, glume, flower and embryo, but not in endosperm". I don't think root is green tissue, you should modify it and similar sentences across the manuscript.

Response: Thank you for the valuable suggestion. The sentence has been modified and pointed change has been incorporated in whole manuscript.

  1. Fig 4: No. 2 lane has multiple bands, you should indicate which band is the target band.

Response: The amplified bands are indicated by red arrows now.

  1. Line 229-230: "Figure 8bB,bD" change to "Figure 8b B&D"; "Figure 8bA,bC" change to " Figure 8b A&C".

Response: The figure citations style has been modified, as suggested.

  1. Line 260: what is the difference of the two "G-Box, G-box"? 

Response: The G-Box and G-box are two different cis elements having different sequences.

  1. Line 283: "genes by interacting with transcription factors such as CCAAT box and MYB". I don't think CCAAT box is a transcription factor. The authors should read the manuscript carefully again and improve the English writing.

Response: Thanks for the correction. The text has been modified (see lines 287-288).

Reviewer 3 Report

Very good job! Well done

Author Response

Thank you so much for appreciation